# How RSV Proteins Join Forces to Overcome the Host Innate Immune Response

**DOI:** 10.3390/v14020419

**Published:** 2022-02-17

**Authors:** Tessa Van Royen, Iebe Rossey, Koen Sedeyn, Bert Schepens, Xavier Saelens

**Affiliations:** 1VIB-UGent Center for Medical Biotechnology, VIB, 9000 Ghent, Belgium; tessa.vanroyen@vib-ugent.be (T.V.R.); iebe.rossey@vib-ugent.be (I.R.); koen.sedeyn@vib-ugent.be (K.S.); bert.schepens@vib-ugent.be (B.S.); 2Department for Biochemistry and Microbiology, Ghent University, 9000 Ghent, Belgium

**Keywords:** respiratory syncytial virus, innate immunity, interferon, non-structural proteins

## Abstract

Respiratory syncytial virus (RSV) is the leading cause of severe acute lower respiratory tract infections in infants worldwide. Although several pattern recognition receptors (PRRs) can sense RSV-derived pathogen-associated molecular patterns (PAMPs), infection with RSV is typically associated with low to undetectable levels of type I interferons (IFNs). Multiple RSV proteins can hinder the host’s innate immune response. The main players are NS1 and NS2 which suppress type I IFN production and signalling in multiple ways. The recruitment of innate immune cells and the production of several cytokines are reduced by RSV G. Next, RSV N can sequester immunostimulatory proteins to inclusion bodies (IBs). N might also facilitate the assembly of a multiprotein complex that is responsible for the negative regulation of innate immune pathways. Furthermore, RSV M modulates the host’s innate immune response. The nuclear accumulation of RSV M has been linked to an impaired host gene transcription, in particular for nuclear-encoded mitochondrial proteins. In addition, RSV M might also directly target mitochondrial proteins which results in a reduced mitochondrion-mediated innate immune recognition of RSV. Lastly, RSV SH might prolong the viral replication in infected cells and influence cytokine production.

## 1. Introduction

Human respiratory syncytial virus (RSV) is an enveloped, negative-stranded RNA virus that is classified in the Pneumoviridae family of the order *Mononegavirales*. Infants, the immunocompromised, and the elderly are at risk to develop severe disease upon RSV infection. RSV is the leading cause of severe acute lower respiratory tract infections (ALRI) in infants worldwide [1]. RSV is estimated to be responsible for about 66,000 to 199,000 deaths annually in children below 5 years of age, with the most fatalities occurring in developing countries. RSV is also responsible for a significant disease burden in young children. In 2015, there were an estimated 3.2 million hospital admissions in this age group due to RSV [1,2]. In addition, there were approximately 500,000 hospital admissions of older adults due to RSV-associated acute respiratory infection in 2015 [3].

The RSV genome is approximately 15,200 nucleotides long. Starting from the 3’ leader region, the genome contains 10 genes, the transcripts of which are translated into 11 viral proteins: non-structural protein 1 (NS1) and 2 (NS2), nucleocapsid protein (N), phosphoprotein (P), matrix protein (M), small hydrophobic protein (SH), attachment glycoprotein (G), fusion protein (F), M2-1 and M2-2 protein (both translated from the M2 transcript that contains two partially overlapping reading frames), and polymerase protein (L) [4] (Figure 1).

In response to virus infection, many elements of the host’s innate immune system are deployed as an early line of defense [5,6]. Pattern recognition receptors (PRRs) can sense viral particles and virus-derived nucleic acids and induce signaling pathways that result in the production of a wide variety of cytokines and chemokines. Attracted by some of these molecules, innate immune cells such as macrophages, monocytes, dendritic cells (DCs), and natural killer (NK) cells are recruited to the site of infection and contribute to the containment of the virus [7]. Activated PRRs trigger several signaling pathways that eventually lead to an upregulated interferon (IFN) gene expression. Type I IFNs (IFN-α and IFN-β) bind to the type I IFN receptor and stimulate the expression of several IFN stimulated genes (ISGs) through JAK/STAT signaling. The 2′-5′ oligoadenylate synthetase (OAS)/RNAse L system, protein kinase R (PKR), IFN-induced tetratricopeptide repeat (IFIT) proteins, and Mx proteins are examples of well-characterized ISGs that possess enzymatic and broad antiviral activities. A more elaborate overview of the antiviral innate immunity can be found in other reviews [6,8]. The quality of the innate immune response is a major determinant for RSV disease severity [9]. Furthermore, the robustness of this initial response also drives the outcome of the subsequent adaptive immune response. Humans, however, can become infected with RSV multiple times throughout life, which means that RSV infection does not elicit lasting protection [10,11]. A better understanding of the immune responses elicited by RSV and the mechanisms exploited by RSV to escape these responses are therefore critical for the development of immune-modulating drugs that may help RSV patients. Indeed, RSV has evolved several mechanisms to circumvent the host’s innate immunity in order to promote virus replication. NS1 and NS2, for example, are well-known antagonists of IFN induction and signaling. Yet it is becoming increasingly clear that NS1 and NS2 are not the only RSV proteins involved in the subversion of the host response. In this review, we highlight and discuss the mechanisms exploited by the different RSV proteins in innate immune evasion.

## 2. Innate Immune Recognition of RSV

PRRs are crucial for the recognition of molecules that are present in pathogens, the so-called pathogen-associated molecular patterns (PAMPs) [12]. PRRs play an indispensable role in the proper function of the host’s innate immunity. Three classes of PRRs are involved in the recognition of RSV PAMPs: Toll-like receptors (TLRs), retinoic-acid-inducible gene-I (RIG-I)-like receptors (RLRs), and nucleotide-binding oligomerization domain (NOD)-like receptors (NLRs). Upon RSV infection, PRRs become activated which leads to the production of cytokines, mainly type I and III IFNs, ISGs, and other mediators [9] (Figure 2).

RIG-I and melanoma differentiation-associated gene 5 (MDA5) are two important RLRs for RSV recognition. Initially, a dominant role for RIG-I was proposed in sensing RSV during the early phases of infection [13,14,15]. Viral mRNA could be co-immunoprecipitated with RIG-I, but not with MDA5, which further confirmed the hypothesis that mainly RIG-I was involved in RSV RNA recognition [13]. In addition, RIG-I knockout mouse fibroblasts were more permissive for RSV infection [14]. Later, it became clear that also MDA-5 plays a significant role in RSV recognition. The group of Lifland et al., reported that MDA5, like RIG-I, colocalizes with RSV genomic RNA, suggesting a previously underappreciated role for MDA5 in RSV sensing [16]. RSV-infected mice deficient in mitochondrial antiviral signaling protein (MAVS), the adaptor protein for both RIG-I and MDA5, showed a strong reduction in type I IFN production [17,18]. These two cytoplasmic RLRs are, however, triggered by different RSV-derived RNA species: RIG-I is activated by 5′ triphosphorylated ss- and dsRNA, whereas MDA5 is activated by long dsRNA molecules independent of their 5′ phosphorylation status [19,20,21].

The NLRs are also involved in host defense against RSV infection, especially through NOD2. The expression levels of NOD2 are enhanced by RSV infection in human peripheral blood mononuclear cells (PBMCs) [22,23]. NOD2 can detect the genomic RSV RNA genome and subsequently activate the innate immune pathway through interaction with MAVS [22]. In addition, NOD2 is involved in the activation of IFN regulatory factor 3 (IRF3) and the production of IFN- β after RSV infection. In line with this, infection of NOD2 knockout mice with RSV resulted in significantly reduced IFN- β levels, higher viral titers, and increased lung pathology compared to wild-type mice [22].

Furthermore, some NLRs are required for activation of the inflammasome upon RSV infection, as was shown for NLRP3 (NOD-like receptor family, pryin domain containing 3) [24]. The NLR inflammasome is a multiprotein complex that includes caspase-1, NLRP3, and the adaptor protein ASC (apoptosis-associated speck-like protein containing a caspase recruitment domain) [25]. This complex promotes the secretion of proinflammatory cytokines (IL-1β and IL-18) and apoptosis through the activation of host caspases [26]. Segovia and colleagues have reported that activation of this NLR inflammasome is critical for interleukin-1β (IL-1β) production during RSV infection [24]. Another report demonstrated that RSV SH might trigger activation of NLRP3 by functioning as a viroporin, leading to disruption of cell membrane integrity [27]. Thus, it appears that SH contributes to the activation of NLRP3 and inflammasome signaling, resulting in the initiation of an antiviral response during RSV infection. In contrast, several groups reported that SH might reduce cytokine release such as IL-1β [28,29,30]. Further validation is required to show an unambiguous effect of SH on NLRP3 activation and subsequent IL-1β release.

In contrast to cytoplasmic RLRs and NLRs, TLRs localize at the cell surface, embedded in the plasma membrane, or intracellularly in the endosomes [31]. In RSV-infected airway epithelial cells, TLR2 expression is upregulated [13,32]. Mice that are deficient in TLR2, which can recognize a wide range of ligands, including molecules with diacyl- and triacylglycerol moieties, proteins, and polysaccharides, or TLR6, which heterodimerizes with TLR2, show strongly increased virus replication and reduced production of proinflammatory cytokines [33]. The mechanism of RSV virion recognition by TLR2/6 is not yet known. TLR3 recognizes dsRNA, which is generated during RSV replication in the cytoplasm as a replication intermediate [34]. Furthermore, TLR3 expression is upregulated upon RSV infection in several cell lines, including A549 and MRC-5 cells [13,34,35]. Like TLR3, TLR7 is present on the endosomal compartment and can sense ssRNA. In RSV-infected TLR7-deficient mice, an increased lung pathology was observed [36], suggesting the involvement of TLR7 in RSV RNA recognition and subsequent innate immune initiation.

The RSV fusion protein can be recognized by TLR4, resulting in activation of nuclear factor-kappa B (NF-κB) signaling [31]. TLR4 is a sensor of bacterial lipopolysaccharides [37]. Upon RSV infection, TLR4 deficient mice showed prolonged viral persistence in the lung when compared to mice expressing TLR4 [38,39]. However, other studies have shown no role for TLR4 in RSV-specific immunity [40,41], leaving the function for TLR4 in recognition and cellular response to RSV uncertain.

## 3. How RSV Proteins Join Forces to Overcome the Host Innate Immunity

Although several PRRs can sense RSV PAMPs and induce IFN production, nasal washes from RSV-infected infants only contain low to undetectable levels of IFN-α and -β in contrast to other respiratory viruses such as influenza A virus and parainfluenza virus [42,43,44]. Furthermore, mononuclear cells from infants with RSV-induced bronchiolitis produce remarkably low levels of IFN-α [45]. RSV infection of human macrophages and PBMCs elicited no detectable levels of type I IFNs [46]. Taken together, these findings suggest that RSV has evolved several mechanisms to prevent type I IFN expression by the infected host cell. The predominant IFN that is induced by nasal epithelium in response to RSV infection is IFN-λ, a type III IFN, which is implicated as the primary IFN to protect airway epithelial cells against respiratory infections [47,48]. RSV can suppress IFN-λ production in lung epithelium [49]. Upon RSV infection, epithelial growth factor receptor (EGFR) is activated and results in increased airway inflammation [50]. RSV-activated EGFR leads to a suppressed IRF-1 induced production of IFN-λ, proposing a mechanism for RSV to escape the host mucosal antiviral response [51].

As outlined below, multiple RSV proteins can hinder the innate immune response of the host against RSV infection. The main players are NS1 and NS2, which have evolved into proteins that have acquired multiple ways to suppress type I IFN production and signaling [52,53] RSV G has a primary function in virion attachment, although its role as an immune modulator should not be underestimated [54]. RSV G reduces the recruitment of innate immune cells by blunting the activity of chemokines and also reduces the production of several cytokines such as IFN-β, IL-10, and IL-12 [55,56,57,58]. RSV N can sequester immunostimulatory proteins to inclusion bodies (IBs) [59,60]. Here, we also propose that RSV N might facilitate the assembly of a multiprotein complex that is responsible for the negative regulation of innate immune pathways. Recently, the nuclear accumulation of RSV M has been linked to an impaired host gene transcription, in particular for nuclear-encoded mitochondrial genes [61]. In addition, there is evidence that RSV M also directly targets mitochondrial proteins to result in a reduced mitochondrion-mediated innate immunity against RSV infection [61,62]. How SH is involved in immune evasion after RSV infection is not yet completely understood, but research suggests that it might prolong the viral replication in infected cells and it can influence cytokine production [28,29,30].

RSV can thus manipulate the immune response through several of its proteins in order to create a favorable environment for its own replication and to make the host more susceptible to (re)infection. Here, we provide an overview of the current knowledge about the mechanisms exploited by RSV proteins in innate immune evasion.

### 3.1. Non-Structural Protein 1 (NS1) and 2 (NS2)

NS1 and NS2 are non-structural proteins, meaning that they are not present in the mature virion [63]. The NS1 and NS2 genes are positioned proximal to the 3′ leader region in the viral genome and are therefore the earliest and most abundantly expressed proteins in infected host cells [64]. The presence of these two non-structural proteins distinguishes RSV from other members of *Mononegavirales*, including the closely related human metapneumovirus (hMPV) [65]. NS1 and NS2 have no sequence homology, except for the DLNP tetrapeptide at their C-terminus [66]. Host microtubule-associated protein 1B (MAP1B) is a common interactor of both NS proteins and this interaction is mediated through this tetrapeptide. For NS2, the interaction with MAP1B seems to be essential to decrease endogenous STAT2 levels [66].

The functions of NS1 and NS2 have been extensively studied, showing their role as type I and III IFN antagonists. This is supported by the observation that human and bovine RSV strains that lack NS1 or NS2 are attenuated both in cell culture and in in vivo models. Infection with NS1/NS2 deletion mutants results in increased IFN-β mRNA levels compared to wild-type RSV-infected cells [49,67,68,69,70,71].

The crystal structure of NS1 was solved by Chatterjee et al., and revealed that NS1 comprises of a β-sandwich flanked by three α-helices [72] (Figure 3). Remarkably, RSV NS1 is a structural paralogue of the N-terminal domain of RSV M, despite the lack of sequence homology between NS1 and M. The α3 helix unique to NS1 is critical for IFN inhibition. Wild-type NS1 can inhibit the activation of the IFN-β promoter during infection with non-NS protein-containing Sendai virus, whereas this effect is impaired by truncation of the NS1 α3 helix. In addition, the α3 helix is involved in the suppression of DC maturation and protein stability.

Recently, the crystal structure of RSV NS2 was also reported [73] (Figure 3). NS2 is composed of four α-helices at the N-terminus followed by a 3-stranded antiparallel β-sheet at the C-terminus. A lack of electron density was observed for both the 22 residues at the N-terminus and the 7 residues at the C-terminus, indicating a high degree of flexibility in these regions. This conformational flexibility could be confirmed by a hydrogen-deuterium exchange assay [73].

The various IFN suppressor functions of NS1 and NS2 have been extensively reviewed recently [52,53]. Here, we summarize how these proteins dampen the host innate immunity and discuss recent findings in more detail.

#### 3.1.1. NS1 and NS2 Impair the Interaction between RLRs and MAVS

Both RSV NS1 and NS2 impair the interaction between RIG-I and MAVS. NS1 does this by interacting with MAVS [74] or by targeting tripartite motif-containing protein 25 (TRIM25), which is responsible for ubiquitination and subsequent activation of RIG-I [75]. NS2 inhibits the interaction of RIG-I with MAVS by binding to the N-terminal CARD domain of RIG-I [73,76]. Further structural analysis revealed that the N-terminus of NS2 is responsible for this interaction. Mutations in the N-terminal α1 helix of NS2 did not only reduce binding to RIG-I but also reduced suppression of IFN-β mRNA levels compared to wild-type NS2. NS2 can also bind the CARD domain of MDA5, an interaction that appears to be enhanced by stimulation with poly (I:C). Activation of RLRs occurs upon RNA binding and is further enhanced by dephosphorylation and ubiquitination. Independent of poly (I:C), NS2 was able to prevent ubiquitination of MDA5 and RIG-I, suggesting that NS2 binds to the inactive RNA-free state of these RLRs. The results above demonstrate that both NS1 and NS2 impair the interaction of the RLRs MDA5 and RIG-I with MAVS and they do so by targeting different signaling molecules (Figure 4).

#### 3.1.2. NS1 and NS2 Suppress Signaling Downstream of MAVS

Downstream of MAVS, the effect of the NS proteins on TRAF3, IKKε, and TBK1 is ambiguous. While some reports demonstrate reduced levels of TRAF3 and IKKε upon NS1 expression [77,78], this could not be confirmed by another group [79]. It was even shown that ectopically expressed NS2 leads to increased levels of both IKKε and TBK1 [76]. In the presence of both NS1 and NS2, IKKε expression is reduced in A549 cells, indicating that the effect of NS2 is inferior to the effect of NS1 [78].

More downstream in the IFN induction pathway, the activation of IRFs and NF-κB is disturbed by the RSV NS proteins. NS1 and NS2 can act cooperatively to disrupt the activation and nuclear shuttling of IRF3 [80]. Ren et al., reported that NS1 can directly interact with both IRF3 and its coactivator CREB-binding protein (CBP), thereby attenuating their interaction [79]. This results in a decreased binding of IRF3 to the IFN-β promoter and hence to reduced IFN-β synthesis. Coincident with the expression of NS proteins in RSV-infected cells, NF-κB is activated [81]. At first sight, this seems counterintuitive since NF-κB is essential for the transcription of type I and III IFNs. However, NF-κB promotes the transcription of anti-apoptotic genes, thus favoring viral replication in infected cells. NS1 and NS2 seem to play a crucial role in the suppression of premature apoptosis through NF-κB by activation of the phosphatidylinositol 3-kinase (PI3K)/protein kinase B (AKT) pathway.

#### 3.1.3. NS1 and NS2 Interfere with JAK/STAT Signaling

NS1 and NS2 also target several proteins downstream of the type I IFN receptor, such as the Janus kinase/signal transducer and activator of transcription (JAK/STAT) signaling pathway. Several reports suggest that NS1 and/or NS2 induce the expression of two suppressor of cytokine signaling proteins (SOCS), being SOCS 1 and SOCS3 [82,83,84]. When upregulated by NS1 and NS2, these proteins show enhanced inhibitory effects on JAK kinases, suppressing subsequent signaling. Moreover, NS1 and NS2 lead to proteasome-mediated degradation of STAT2, further reducing the IFN-induced signaling [84,85,86] (Figure 4).

#### 3.1.4. NS1 and NS2 Are Part of a Large Degradative Complex

It is remarkable how NS1 and NS2 can affect such a broad range of proteins that are important for the host antiviral immune response. Goswami et al., proposed the existence of an NS degradasome (NSD) complex that comprises NS1 and NS2 together with several host factors [77]. The precise composition of this NSD still needs to be determined. The NSD might be involved in the degradation of multiple key IFN signaling molecules such as RIG-I, TRAF3, IKKε, IRF3, IRF7, and STAT2. The degradative activity of this complex is strongly dependent on both mitochondrial association and MAVS. The exact mechanism of this proteasomal degradation driven by the NS proteins is currently not clear but is thought to be linked to the ability of NS1 to interact with elongin C and cullin-2 [85]. This allows NS1 to act as a scaffold on which an E3 ligase complex can be composed. Although NS2 lacks cullin-2 and elongin C interaction sites, NS2 can induce ubiquitination of STAT2 [87]. Altogether, these findings suggest that the NS proteins can act as inducers of host protein ubiquitination, followed by NSD-mediated degradation.

#### 3.1.5. Nuclear NS1 Modulates ISG Transcription

In addition to its cytoplasmic localization, several reports have demonstrated NS1 partitioning into the nucleus [66,73,88,89,90,91]. However, these findings are based on overexpression of NS1. Recently, Pei and colleagues showed that NS1 is actively transported into the nucleus upon RSV infection in primary human lung epithelial cells, probably through interaction with the nuclear transporter exportin-1 (XPO1) [92]. In addition, nuclear NS1 colocalizes with the Mediator complex and associates with chromatin, suggesting a potential role in transcription regulation. The interaction of NS1 with the Mediator subunits MED1, MED14, and MED25 was confirmed by co-immunoprecipitation (co-IP). The Mediator complex conveys signals between transcription factors and the RNA polymerase II machinery, resulting in transcriptional regulation of specific genes, including antiviral genes [93,94]. Chromatin immunoprecipitation experiments revealed enrichment of NS1 within regulatory elements of differentially expressed genes during RSV infection, including immune response genes (e.g., IFIT2, IFIT3, OAS1, …) [92]. This study demonstrated that NS1 does not only bind to regulatory elements in the IFIT locus but is also responsible for suppressed transcription of these ISGs. These results are in agreement with other reports demonstrating that ectopic expression of NS1 results in reduced protein levels of the ISGs 2′-5′ oligoadenylate synthetase-like protein (OASL), IFIT1, and interferon-induced transmembrane protein 3 (IFITM3) [95,96]. Although a proteasome-dependent mechanism was initially proposed for this observation, it cannot be excluded that the nuclear presence of NS1, where it can disturb transcription of these ISGs, is also partly responsible for this effect. It would be interesting to investigate how RSV replication is altered upon suppression of Mediator subunits to gain a better understanding of the relevance of the interaction between NS1 and Mediator subunits.

Taken together, Pei and colleagues showed that nuclear localization of NS1 might aid to disrupt host responses against RSV infection through association with chromatin at promoters and enhancers of immune response genes [92]. These findings revealed yet another mechanism of how NS1 counteracts host antiviral mechanisms during RSV infection.

In conclusion, NS1 and NS2 work cooperatively to suppress the host’s innate immune responses by targeting several signaling molecules involved in IFN induction and IFN response pathways (Figure 4). In the nucleus, each pathway converges on a certain transcription factor that might be involved in the recruitment of the Mediator complex. This complex can be regarded as an endpoint of a set of signaling cascades, given its role in relaying signals of TFs to the transcription machinery [97]. Eventually, this leads to transcription factor-dependent activation (or suppression) of specific genes involved in the response to the stimulus. Since nuclear NS1 targets the Mediator complex, this might be a very efficient way to outsmart the host response to an RSV infection.

**Figure 4 viruses-14-00419-f004:**
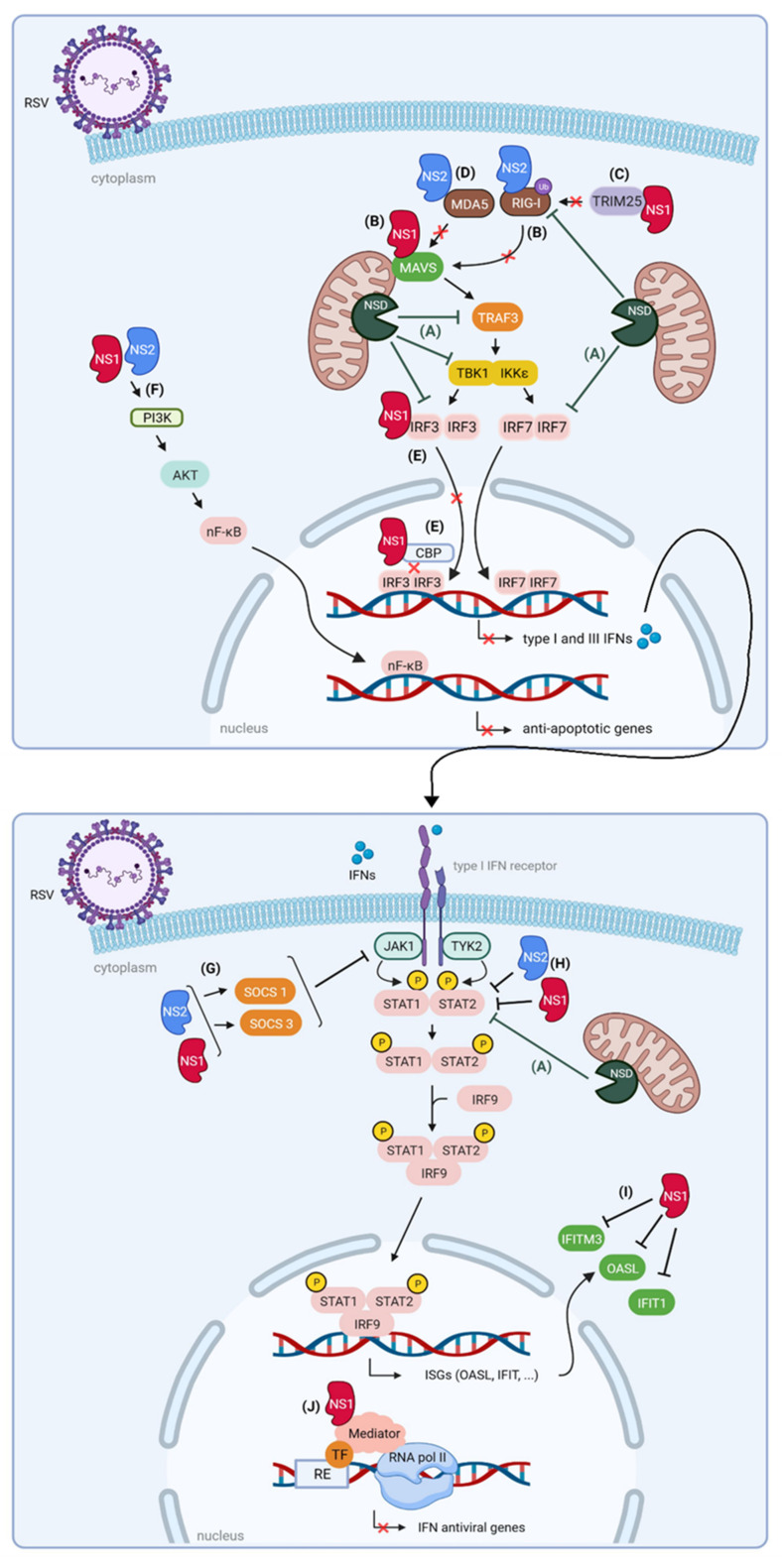
Proposed roles of RSV NS proteins in suppression of host innate immune responses. NS1 and NS2 can target different proteins that are necessary for the host antiviral immune response. (**A**) The existence of an NS degradasome (NSD) complex encompassing NS1 and NS2 together with several host factors in RSV-infected cells has been proposed. The NSD might be involved in the degradation of RIG-I, TRAF3, IKKε, IRF3, IRF7, and STAT2. (**B**) The binding of RIG-I to MAVS is abrogated by both NS proteins, where NS1 interacts with MAVS and NS2 targets RIG-I. This results in a cooperative suppression of downstream MAVS signaling. (**C**) NS1 interacts with TRIM25, which is responsible for ubiquitination and subsequent activation of RIG-I. (**D**) NS2 binds to MDA5, which is then less available for binding to MAVS. (**E**) NS1 can directly interact with both IRF3 and its coactivator CBP, attenuating the interaction between the latter two. This results in decreased recruitment of IRF3 to the IFN-β promoter, inhibiting IFN-β synthesis. In addition, NS1 and NS2 might also disrupt the activation and nuclear shuttling of IRF3. In combination with NSD targeting IRF7, these observations all result in decreased transcription of type I and III IFNs. (**F**) NS1 and NS2 activate the PI3K/AKT pathway, subsequently leading to NF-κB activation. NF-κB is involved in the transcription of anti-apoptotic genes. (**G**) NS1 and NS2 induce expression of SOCS1 and SOCS3, leading to inhibition of JAK activity and downstream signaling. (**H**) Decreased STAT2 expression levels are observed upon expression of NS1 and NS2. In addition, STAT2 seems to be a substrate of the NSD complex, leading to impaired JAK/STAT signaling and reduced transcription of ISGs. (**I**) Ectopic expression of NS1 results in reduced protein levels of the antiviral ISG products OASL, IFIT1, and IFITM3. (**J**) Nuclear NS1 colocalizes with the Mediator complex and associates with chromatin, resulting in suppressed transcription of IFN-induced antiviral genes. AKT, protein kinase B; CBP, CREB-binding protein; NSD, NS degradasome complex; IFN, interferon; IFIT1, interferon-induced protein with tetratricopeptide repeats 1; IFITM3, interferon-induced transmembrane protein 3; IRF3, interferon regulatory factor 3; IRF7, interferon regulatory factor 7; IRF9, interferon regulatory factor 9; ISG, interferon-stimulated gene; JAK1, Janus kinase 1; MAVS, mitochondrial antiviral-signaling protein; MDA5, melanoma differentiation-associated protein 5; NF-κB, nuclear factor-kappa B; OASL, 2′-5′ oligoadenylate synthetase-like protein; PI3K, phosphatidylinositol 3-kinase; RE, regulatory element; RIG-I, retinoic-acid-inducible gene-I; RNA pol II; RNA polymerase II; RSV, respiratory syncytial virus; STAT1/2, signal transducer and activator of transcription TBK1, tank binding kinase 1; TF, transcription factor; TRAF3, tumor necrosis factor receptor-associated factor 3; TRIM25, tripartite motif-containing protein 25; and TYK2, tyrosine kinase 2. Figure created with BioRender software (BioRender.com (accessed on 13 January 2022)).

### 3.2. Nucleoprotein (N)

The RSV nucleoprotein (N) forms decameric rings around which the viral RNA genome is tightly wrapped, resulting in left-handed helical nucleocapsid structures [98,99].

The nucleocapsid serves as a template for both RNA transcription and replication, which are catalysed by the viral RNA-dependent RNA polymerase. By encapsidating the viral genome and antigenome, RSV N impairs recognition by cellular PRRs and prevents degradation of the viral RNA.

#### 3.2.1. RSV N Sequesters Innate Immune Signaling Proteins to Inclusion Bodies

RSV N also counteracts innate immune recognition by sequestering immunostimulatory proteins to so-called IBs, which are induced by N and P expression [59,60]. These IBs are cytoplasmic structures that are considered as sites of viral replication and transcription, where all viral proteins of the polymerase complex are concentrated [60]. Early in RSV infection, MAVS and MDA5 are sequestered to small IBs [16] (Figure 5). This sequestration was also observed when N was expressed alone or simultaneously with P in Vero cells, suggesting that RSV N is responsible for localizing MAVS and MDA5 within IBs. In addition, RSV N has been shown to co-immunoprecipitate with ectopically expressed MDA5, but not with MAVS [16]. It is possible that N interacts with MAVS nevertheless, but that this interaction does not withstand the lysis conditions that have been used in the co-IP experiments. However, colocalization between RSV N and both cellular proteins could be demonstrated by a proximity ligation assay [16]. Other lysis-free methods, for example, based on virotrap or proximity-dependent biotin identification, could, perhaps, substantiate an interaction between MAVS and RSV N [100,101].

#### 3.2.2. RSV N Interacts with PKR

N can interact with PKR and thereby blunts the downstream signalling of this IFN-induced cytoplasmic serine-threonine kinase [102,103]. Activated PKR halts protein synthesis by phosporylation of the alpha subunit of the eukaryotic translation initiation factor eIF2α. PKR activation, through increased phosphorylation, by RSV infection has been described [34,102]. However, no increased phosphorylation of eIF2α is observed during RSV infection and thus mRNA translation is maintained. Instead, it was shown that the binding of RSV N to PKR prevents the association of PKR with its canonical substrate eIF2α. In addition to the well-characterized translation inhibition function, PKR is an essential component of innate immune signaling [104]. Activated PKR mediates the activation of mitogen-activated protein kinase (MAPK), the IκB kinase (IKK) complex, and MAVS. Subsequently, this leads to downstream activation of respectively c-Jun/ATF2, NF-κB, and IRF3, resulting in transcription of cytokines, including IFNs (Figure 5). Besides, it has been reported that RSV infection results in activation of both MAPK and NF-κB pathways [105]. Considerably less activation of these factors was observed in the lungs of RSV-infected PKR knockout compared to wild-type mice, which suggests that PKR is important for optimal activation of these signaling pathways in vivo in response to RSV infection. As binding of N with PKR limits eIF2α phosphorylation, this interaction might also reduce PKR-mediated downstream signaling through MAPK and IKK. This would provide new insights into the role of RSV N as an innate immune suppressor, although further research is necessary to confirm this hypothesis.

#### 3.2.3. RSV N Might Downregulate NF-κB and IRF3/7 Signaling

Another immunomodulatory mechanism is related to the interaction of N with host Tax1-binding protein 1 (TAX1BP1). This interaction was recently identified in a yeast two-hybrid screen using RSV N as a bait [106]. The interaction was further validated by GST-pulldown and by co-IP in BSRT7 cells. In TAX1BP1 knockout mice, RSV replication was strongly decreased compared to wild-type mice. In parallel, higher levels of IFN-α and TNF-α were detected in the lungs of TAX1BP1 knockout mice compared to wild-type mice. Infection of alveolar macrophages isolated from TAX1BP1 knockout mice resulted in enhanced secretion of antiviral and inflammatory cytokines compared to RSV-infected macrophages isolated from wild-type mice [106]. These findings suggest a proviral role of TAX1BP1 during RSV replication by participating in the attenuation of the host antiviral and inflammatory response. This is in line with another report that showed an increased inflammatory and antiviral response in RSV-infected A549 cells after silencing of TAX1BP1 [107].

TAX1BP1 acts as a cofactor of the ubiquitin-editing enzyme A20, which is responsible for negative regulation of the NF-κB and IRF3/7 pathways by targeting several molecules of the RIG-I pathway for ubiquitination/deubiquitination [107]. RSV PAMPs result in activation of host PRRs, eventually leading to the expression of inflammatory and antiviral genes through activation of NF-κB and IRF3/7. It is crucial that these signaling pathways are tightly regulated to avoid excessive tissue damage and inflammatory responses while ensuring viral clearance. Post-translational modifications such as ubiquitination/deubiquitination contribute to fine-tuning of this immune response [108,109]. The inhibitory functions of A20 are dependent on the formation of a multiprotein complex that includes TAX1BP1, A20-binding inhibitor of NF-κB1 (ABIN1), and the ubiquitin ligases ITCH and RING finger protein 11 (RNF11) [110]. It has been demonstrated that downregulation of TAX1BP1, A20, or ABIN1 leads to enhanced levels of different cytokines and chemokines in RSV-infected cells [107]. However, only several of the tested genes were upregulated in ITCH-silenced cells and none in RNF11-silenced cells. These results suggest that A20 requires at least two other proteins (TAX1BP1 and ABIN1) to interrupt the innate immune response in RSV-infected epithelial cells. It has been demonstrated that ABIN1 recruits TAX1BP1 and A20 to TBK1 and IKKε in response to poly (I:C) treatment [111]. TAX1BP1 then cooperates with ABIN1 to disrupt the interaction between TRAF3 and TBK1/IKKε.

Based on this clarified mechanism of the A20 protein complex [111] and the recent findings that RSV N interacts with TAX1BP1 [106], we could hypothesize that RSV N facilitates the assembly of this multiprotein complex (Figure 5). In this way, RSV N would be able to restrain the innate immune response during infection. To further confirm this hypothesis, it would be interesting to discover the residues of RSV N that are responsible for the interaction with TAX1BP1. Subsequently, the assembly of the A20 protein complex in cells infected with wild-type RSV can be compared to infection with RSV where N is impaired in TAX1BP1 binding. In addition, other proteins (apart from A20, TAX1BP1, and ABIN1) may participate in this complex required for innate immune suppression against RSV infection.

### 3.3. Matrix (M) Protein

The RSV Matrix protein is a structural protein located below the lipid envelope, that forms a matrix layer around the nucleocapsid [112]. By bridging the viral envelope and the nucleocapsid, M plays a central role in virus assembly.

Early in infection (6–10 h post-infection), M translocates to the nucleus via the nuclear pore complex. This is mediated by the nuclear transport protein importin β1 that recognizes a nuclear localization signal in M [113]. Once in the nucleus, RSV M can inhibit host cell transcription [61]. Later in infection, M shuttles back to the cytoplasm via exportin 1, where it associates with IBs [114]. Here, M is hypothesized to mediate the transport of newly synthesized ribonucleoproteins to the cell surface to allow the assembly of new virions [115]. In addition, M has been shown to promote elongation and maturation of virus filaments, although it is not required for the initiation of filament formation [116]. Crystallography studies revealed that dimerization and oligomerization of RSV M is necessary to drive viral budding and release, similar to what is described for paramyxovirus M proteins [117,118].

#### 3.3.1. Nuclear M Downregulates Transcription of Nuclear-Encoded Mitochondrial Genes

RSV infection induces global changes in the host transcriptome and proteome [92,119,120,121,122]. Several arrays indicate that the expression of a substantial amount of cellular genes is altered after infection with RSV [123,124,125,126]. As discussed above, nuclear-localized RSV NS1 modulates host gene transcription and might be, at least partially, responsible for this observed effect. Recently, Li et al., demonstrated that nuclear accumulation of RSV M early during infection is also associated with impaired host cell transcription [61]. This inhibition of the host cell transcription is mediated at least in part by the RSV M protein via a pair of basic amino acid residues (R170/K172) in the central RNA binding domain. Although R170/K172 is not involved in RNA binding, these residues are critical for DNA binding in vitro and for chromatin binding in living cells. Recombinant RSV, in which these two critical amino acids are substituted (rA2M/R170T:K172T), displayed delayed and decreased viral replication in several cell lines. The attenuated replication of this mutant virus in A549 cells was concomitant with loss of inhibition of host cell transcription, which was observed in cells infected with wild-type RSV. Interestingly, after infection with the mutant RSV virus, the observed increase in gene expression was in particular pronounced for genes encoding mitochondrial proteins (TOM40, SLC25A1), when compared to infection with wild-type RSV. These results suggest that the binding of RSV M to host chromatin is important to favor virus production through inhibition of host cell transcription with a specific effect observed for mitochondrial genes. It is not yet described whether also mitoprotein levels and mitochondrial function are affected by RSV infection. Interestingly, however, the RSV M protein can be detected in the mitochondrial fraction of RSV-infected A549 cells [121], which further suggests that M might be involved in altering the mitochondrial function upon RSV infection. Since accurate functioning of the mitochondria is essential to control both innate and adaptive immune responses [127], RSV M might disturb the host immune mechanisms by impairment of the mitochondrial function.

#### 3.3.2. RSV M Interacts with a ZNF Protein

Another link between RSV M and the innate immune system has been found after a high-throughput microfluidics-based affinity screen using M as a prey [62]. A custom library of 500 human proteins was expressed, immobilized on a microfluidic chip, and tested for the binding to recombinant RSV M protein. This screen resulted in the identification of 24 novel host factors that can directly interact with the M protein. Several of these identified host proteins might be involved in innate immunity signaling, including DNA binding Zinc-finger proteins (ZNFs). The interaction of RSV M with ZNF502, one of the eight ZNFs identified in the screen, was further validated by co-IP. Knockdown of this host protein in HEK293T cells resulted in the reduction of infectious RSV release. ZNF proteins play a diverse role in transcription regulation by recruitment of transcription co-activators/co-repressors, chromatin modifiers, and other transcription factors, resulting in the activation or suppression of downstream genes [128]. Interestingly, M showed nuclear accumulation and decreased viral filament formation upon ectopic expression of ZNF502 [62]. It is tempting to hypothesize that, in addition to RSV M association with chromatin, the binding of M to ZNF proteins is another strategy of M to hijack host transcription. Further work is required to support this hypothesis and to investigate whether this interaction also impacts mitochondrial genes in particular, as is observed for RSV M association with chromatin.

#### 3.3.3. RSV M Interacts with Mitochondrial Proteins That Mediate Host Innate Immune Responses

The screen performed by Kipper et al., also identified mitochondrial proteins involved in the innate immune response as potential M interacting factors [62]. These include the Voltage-dependent anion-selective channel protein 1 (VDAC1), which is believed to be involved in mitochondria-mediated apoptosis [129], and the translocase of outer mitochondrial membrane 22 (TOM22). The TOM complex allows the import of nuclear-encoded preproteins into mitochondria, with TOM22 being the major site for preprotein binding [130]. Subunits of the TOM complex also participate in interactions that promote the host’s innate immune responses after viral infection [121,130]. The interaction between RSV M and TOM22 was confirmed by co-IP [62].

The group of Munday et al., showed that TOM70, another subunit of the TOM complex, acts in an antiviral manner during RSV infection [121]. In addition, this group observed an RSV-induced mitochondrial association of several innate immune proteins. In Sendai virus (SeV), vesicular stomatitis virus (VSV), and Newcastle disease virus (NDV) infected cells, TOM70 is involved in the recruitment of mediators of innate immunity to the mitochondria [131,132]. Based on these findings and the observation that mitochondrial import mechanisms are altered in RSV-infected cells, the group of Munday et al., Hypothesized that TOM70 might also attract IFN-regulated proteins to mitochondria in RSV-infected cells [121]. In this way, TOM70 favors innate immune signaling, explaining its observed antiviral effect during RSV infection.

Based on these studies and as illustrated in Figure 6, we hypothesize that M mainly targets mitochondrial proteins to overcome the host innate immunity against RSV infection. It is tempting to speculate that RSV M boycots the function of TOM subunits to impair the mitochondrion-mediated innate immune response. M might do so by either interacting with TOM subunits directly (as was shown for TOM22) or by inhibiting their transcription (as was shown for TOM40) [61,62]. The TOM complex is embedded in the outer mitochondrial membrane. Surveillance of the functionality of these outer membrane proteins is crucial for the function of the mitochondria, which play a key role in the formation of a signaling platform involved in innate immune signaling [133]. It still needs to be demonstrated whether the interaction of RSV M with TOM22 or the impact of RSV M on TOM40 gene expression can subvert TOM-mediated antiviral activity in response to RSV infection.

The inhibitory effect of nuclear RSV M on host gene transcription, with nuclear-encoded mitochondrial genes, in particular, can be attributed to the binding of RSV M with chromatin [61]. It would be of interest to investigate whether the interaction of RSV M with ZNF proteins, as demonstrated by Kipper et al. [62], also contributes to this observed effect.

### 3.4. Small Hydrophobic (SH) Protein

The small hydrophobic (SH) protein of RSV is a type II transmembrane protein, present on the RSV surface. SH deleted RSV mutants replicate as efficiently as wild-type RSV in cell culture. In mice, however, RSV from which the SH gene is deleted (RSV∆SH) seems less virulent than wild-type RSV [134,135,136]. Structural studies propose that SH acts as a viroporin and forms pentameric ion channels [137,138]. Although the function of SH in the RSV replication cycle is not yet completely understood, several studies have provided evidence of a potential immunomodulatory role for SH. First, RSV∆SH shows enhanced TNF signaling and increased production of IL-1β [28,29,30]. In contrast, another report demonstrated that SH is involved in the activation of the NLRP3 inflammasome, which results in the secretion of IL-1β [27]. Further validation is necessary to clarify these contradictory effects of SH on IL-1 β production; however, more evidence is present for a suppressive effect of SH on IL-1β after RSV infection. A second immunomodulatory mechanism that is exerted by SH is its involvement in phosphorylation of NF-κB p65, which is a crucial step for the regulation of proinflammatory cytokines [30]. Infection of antigen-presenting cells (APCs) with bovine RSV resulted in inhibition of NF-κB p65 phosphorylation and a reduction in proinflammatory cytokine levels. Interestingly, this effect was not observed when an SH deleted virus was used for infection. This indicates that SH might modulate the function of APCs by the suppression of the NF-κB pathway resulting in decreased proinflammatory cytokine production and subsequently reduced activation of T cells. Finally, as was demonstrated for SH of parainfluenza virus type 5, RSV SH can delay apoptosis in infected cell cultures [28]. SH interacts with B-cell receptor-associated protein 31 (BAP31), an apoptosis regulator [139].

### 3.5. Glycoprotein (G)

The RSV glycoprotein (G) is a type II membrane protein that mediates attachment of the virion to the host cell [54]. More specifically, RSV G targets ciliated human airway epithelial cells (HAE) [140]. Besides the full-length membrane-anchored G protein (mG), an N-terminally truncated soluble form of G is secreted (sG) by RSV-infected cells [141]. The synthesis of sG is the result of translation initiation at the second, in frame, start codon that is located in the transmembrane-coding region [142] (Figure 7). sG is already detectable at 6 hpi in HEp-2 cells, whereas mG is only detectable at later time points [143]. This suggests that sG might play a role during the early antiviral response.

mG consists of three domains: the cytoplasmic domain, the transmembrane domain, and the ectodomain [140] (Figure 7). The ectodomain comprises two hypervariable, mucin-like regions that are glycosylated with O-glycans and N-glycans [144]. RSV G is regarded as the most variable protein, with only 53% identity between subgroups A and B [144]. In addition, the G ectodomain includes a central region that consists of 13 highly conserved amino acids, a cysteine noose, a CX3C motif, and a heparin-binding domain (HBD). The HBD can interact with heparan sulfate on immortalized stable cell lines [145]. Remarkably, heparan sulfate is absent in well-differentiated HAE cells and therefore unlikely to be the receptor for RSV in vivo [146]. RSV G, through its CX3C motif, can also bind to the fractalkine receptor CX3CR1, which is expressed in ciliated cells in HAE cultures [147,148]. These observations provide evidence that RSV probably attaches to the CXC3R1 receptor via the CX3C motif in G in vivo. In addition, CX3CR1 is also expressed by most immune effector cells, such as monocytes, DCs, NK cells, and T lymphocytes [149,150,151]. The interaction between G’s CX3C and the host CX3CR1 affects both innate and adaptive immune responses, which will be elaborated in more detail below. Similar to mG, sG can also bind CXC3R1 through its conserved CX3C motif [152]. Several reports have highlighted that G, in addition to its attachment function, may contribute to RSV evasion of host immunity in a variety of ways [55,56,57,58,152,153,154].

#### 3.5.1. RSV G Modulates Recruitment of Innate Immune Cells by Sequestering Chemokines

An early report by Tripp et al., revealed that an RSV mutant that is deficient in G and SH induced an increased influx of NK cells and neutrophils in the lung of infected BALB/c mice when compared to infection with wild-type RSV [155]. This increased influx was associated with higher levels of chemokines observed in the absence of G and SH [153]. This suggests a potential role of G and/or SH in the suppression of chemokine expression upon RSV infection, resulting in a decreased infiltration of innate immune cells to infection sites.

Later, it was demonstrated that sG affects the migration of immune cells by altering chemokine expression. To our knowledge, it has not yet been evaluated whether SH, as initially presumed, is also involved in this inhibitory effect. In A549 cells infected with RSV∆sG, increased mRNA and protein levels of intercellular adhesion molecule-1 (ICAM-1), IL-8 and Regulated upon Activation, Normal T Cell Expressed and Presumably Secreted (RANTES) were observed, compared to infection with wild-type RSV [55]. RANTES is a chemoattractant for monocytes, basophils, and eosinophils, whereas IL-8 and ICAM-1 have mainly chemotactic activity for neutrophils [156,157,158]. It has already been reported that upon RSV infection, increased expression of ICAM-1, IL-8, and RANTES occurs through activation of NF-κB [159,160,161]. Further experiments revealed that sG exerts inhibitory effects on NF-κB activity, providing a mechanism for the observed reduced expression levels of chemokines upon infection with RSV compared to RSV∆sG. Expression of ICAM-1 on human epithelial cells following RSV infection results in increased adhesion of neutrophils, which in turn contributes to cell damage and death [158]. Pulmonary inflammation with an increased influx of neutrophils after RSV infection was also observed in a murine model [162]. The infection of mice with RSV∆sG resulted in increased infiltration of pulmonary macrophages and eosinophils compared to infection with wild-type RSV, further suggesting that sG reduces infiltration of immune cells to the site of infection. Furthermore, compromised airway function and enhanced airway inflammation were observed after RSV∆sG infection. Together with the abovementioned study, this confirms that RSV sG can modulate the recruitment of innate immune cells both in vivo and in cell culture, thereby protecting the infected cells against excessive damage and thus increasing RSV propagation.

In conclusion, through the expression of sG, RSV can blunt the activity of certain chemokines such as ICAM-1, IL-8, and RANTES, leading to a decreased recruitment of innate immune cells to the site of infection. One potential outcome of this activity of sG is a reduced NF-κB activity. In addition, the presence of the CX3C motif in sG allows binding to CX3CR1 expressed on immune cells [152]. sG thus mimics the chemokine fractalkine [152]. Presumably, there is a competition between fractalkine and RSV sG for binding to CXC3R, allowing RSV sG to subvert the fractalkine-mediated cell trafficking. sG expression thereby dampens the inflammatory response evoked by the infected cells.

#### 3.5.2. RSV G Modulates Cytokine Production

The central conserved region of RSV G shares structural homology with the fourth subdomain of the TNF receptor 1 [163]. RSV-infected PBMCs that were pretreated with RSV G protein purified from the culture supernatant of RSV-infected cells resulted in a suppressed release of TNF, IL-10, and IL-12 compared to infected PBMCs without prior RSV G stimulation [56]. Polack et al., demonstrated that the CXC3 motif of RSV G can inhibit F-mediated production of inflammatory cytokines by isolated human monocytes [57]. Moreover, inflammatory responses elicited by TLR4 agonists other than the RSV F protein could be modulated by this viral protein. Furthermore, infection of monocytes with RSV∆G resulted in increased production of IL-6 and IL-1β compared to wild-type RSV. This effect can be assigned to RSV G’s ability to decrease nuclear translocation of NF-κB. To further confirm these results in vivo, mice were infected with wild-type RSV and RSV∆G. IL-6 concentrations were measured in pulmonary macrophages by flow cytometry, showing an increased production of this cytokine in RSV∆G compared to wild-type RSV. This suggests that RSV G contributes to the control of IL-6 expression by pulmonary macrophages in vivo.

In line with this, it has been demonstrated that ectopically expressed RSV G or sG could inhibit the TLR3/4-mediated IFN response in HEK293 cells [58]. Furthermore, sG is a potent inhibitor of IFN-β production in monocyte-derived DCs in response to poly (I:C). This effect was dependent on Toll-interleukin receptor domain (TIR)-containing adaptor molecule (TICAM-1), a common adaptor for both TLR3 and TLR4.

Additional evidence that RSV G is involved in suppression of the antiviral IFN responses was provided by Moore et al. [82] They reported that type II alveolar cells infected with RSV∆G resulted in significantly higher expression of IFN-β and interferon-stimulated gene 15 (ISG15) compared to wild-type RSV-infected cells. Furthermore, RSV G modulates the expression of SOCS3, which negatively regulates type I IFN signaling, proposing another mechanism of G to affect the host type I IFN response [82].

Chirkova and colleagues showed that infection of A549 cells with a mutant RSV, containing G with an altered CX3C motif, induced higher levels of type I and III IFN compared to infection with wild-type RSV [164]. Inoculation of human DCs and monocytes with the mutant virus resulted in increased production of TNF and IFN-α compared to wild-type RSV. This highlights that the CX3C motif of G plays a significant role in the interference with cytokine production. This could be explained by the binding of G CX3C to the fractalkine receptor CX3CR1, which appears to play a role in PRR-mediated innate immune responses. Mice deficient in CX3CR1 show decreased expression levels of TLR4 and suppressed production of TNF and IFN-γ by macrophages compared to wild-type mice [165,166]. However, the exact mechanism of the G CX3C-induced impairment of the host’s immune response remains unknown.

#### 3.5.3. RSV G Modulates the Function of Innate Immune Cells

Johnson and colleagues demonstrated that RSV G interacts with the lectins DC-SIGN and L-SIGN on DCs [154]. This interaction led to phosphorylation of MAP kinases ERK1 and ERK2, resulting in inhibition of DC activation. In RSV-infected DCs, neutralization of DC/L-SIGN enhanced maturation of DCs and both cytokine (IFN-α) and chemokine production, although only moderate changes were observed. Thus, the interaction of RSV G with DC/L-SIGN on the surfaces of DCs inhibits effects of DC activation, such as cytokine production.

## 4. Concluding Remarks

RSV is a major respiratory pathogen that has evolved many strategies to blunt the host antiviral response. This may help explain why RSV typically causes only mild respiratory symptoms. Still, for the very young and the elderly, RSV infections can result in life-threatening lung disease characterized by immune dysregulation [167]. With more than half of its proteins having an immunomodulating effect, RSV is heavily armored against the host’s innate immune response. This helps to explain the success of the virus and its capacity to infect people throughout life. A surprising feature of RSV, which replicates in the cytoplasm, is the impact on host cell nuclear transcription by NS1 and M. A more clear understanding of the mechanisms that RSV has evolved to circumvent innate immune responses from its host, may form the basis for the development of interventions that, in turn, counteract the immune-suppressing mechanisms of this virus.

## Figures and Tables

**Figure 1 viruses-14-00419-f001:**
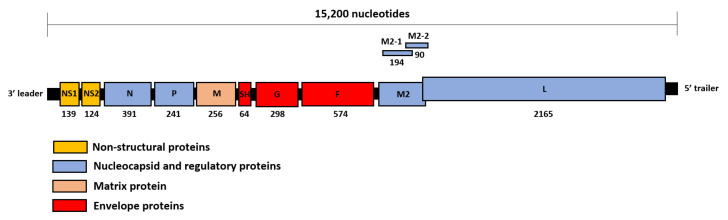
Schematic diagram of the RSV genome organization. The approximate length of the RSV genome is indicated on top of the figure. The number of amino acid residues of the viral proteins is shown below each open reading frame as present on the antisense RSV genome.

**Figure 2 viruses-14-00419-f002:**
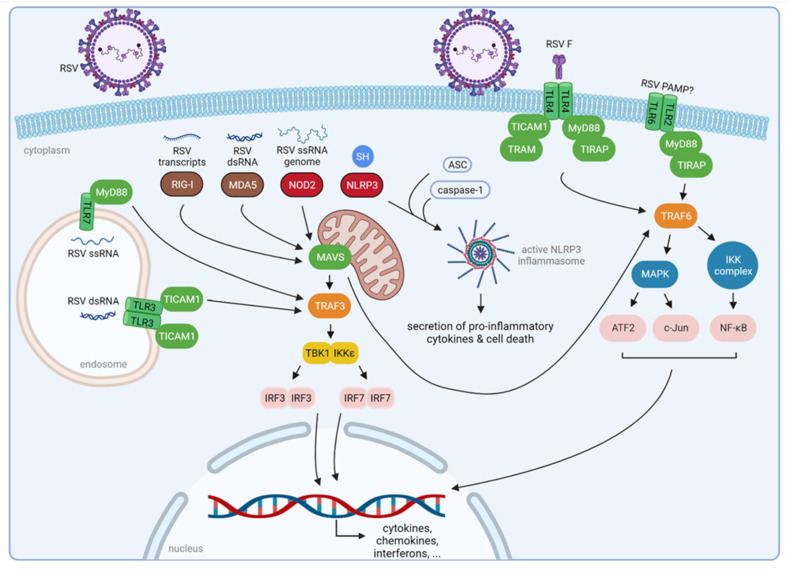
Innate immune recognition of RSV. Three classes of PRRs are involved in the recognition of RSV: TLRs, RLRs, and NLRs. Considering the RLRs, RIG-I binds RSV transcripts, and MDA5 is triggered by RSV dsRNA. The NLR NOD2 recognizes the RSV ssRNA genome. Activation of these RLRs and NOD2 results in downstream activation of mitochondrial-associated MAVS, which in turn induces TRAF3 and TRAF6. The TRAF3 adaptor subsequently activates the two kinases TBK1 and IKKε, resulting in phosphorylation and activation of the transcription factors IRF3 and IRF7. TRAF6 activates the MAPK and the IKK kinase complex, resulting in, respectively, phosphorylation of ATF2/c-Jun and NF-κB. These transcription factors then translocate to the nucleus, inducing the transcription of cytokines, such as interferons, and chemokines, such as RANTES. The NLRP3 inflammasome can be activated by RSV SH. After recruitment of other host proteins, an active NLRP3 inflammasome is assembled, resulting in cell death and secretion of proinflammatory cytokines. TLRs are localized on the cell surface or intracellular components such as endosomes. The mechanism of RSV virion recognition by TLR2/6 remains unknown, whereas TLR3 and TLR7 recognize an RSV dsRNA intermediate and ssRNA, respectively. TLR4 is suggested to be activated by RSV F. Activation of TLRs results in the recruitment of adaptor proteins necessary for downstream signaling. TLR2 recruits MyD88 and TIRAP, as does TLR4. In addition, TLR4 can recruit TICAM1 and TRAM. MyD88 is used by TLR7 as an adaptor protein, whereas TLR3 signals through TICAM1. The adaptor proteins converge at TRAF3 or TRAF6 which become activated, ultimately resulting in an antiviral response through downstream signaling. ASC, apoptosis-associated speck-like protein containing a caspase recruitment domain; ATF2, activating transcription factor 2; IKK, inhibitor of nuclear factor kappa-B kinase; IRF3, interferon regulatory factor 3; IRF7, interferon regulatory factor 7; MAPK, mitogen-activated protein kinase; MAVS, mitochondrial antiviral-signaling protein; MDA5, melanoma differentiation-associated protein 5; MyD88, myeloid differentiation primary response protein MyD88; NF-κB, nuclear factor-kappa B; NLR, nucleotide-binding oligomerization domain (NOD)-like receptors; NLRP3, NOD-like receptor family, pryin domain containing; NOD2, nucleotide-binding oligomerization domain-containing protein 2; PAMP, pathogen-associated molecular pattern; PRR, pattern recognition receptor; RIG-I, retinoic-acid-inducible gene-I; RLR, RIG-I-like receptor; RSV, respiratory syncytial virus; TBK1, tank binding kinase 1; TICAM1, toll/interleukin-1 receptor domain-containing adapter molecule 1; TIRAP, toll/interleukin-1 receptor domain-containing adapter protein; TLR, toll-like receptor; TRAF3, tumor necrosis factor receptor-associated factor 3; TRAF6, tumor necrosis factor receptor-associated factor 6; and TRAM, toll-like receptor adaptor molecule. Figure created with BioRender software (BioRender.com (accessed on 13 January 2022)).

**Figure 3 viruses-14-00419-f003:**
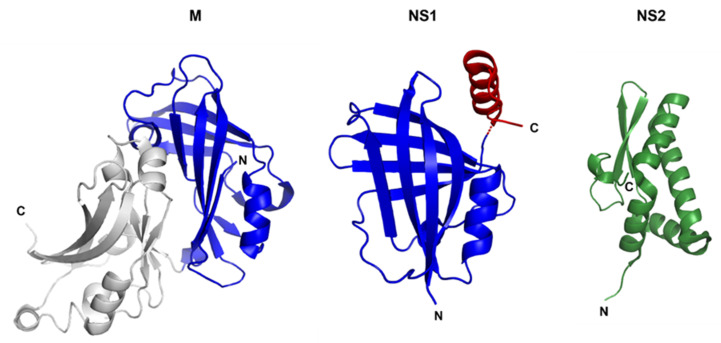
Cartoon representations of the crystal structure of RSV M, NS1, and NS2. The N-terminal domain of RSV M shows structural similarities to NS1 (blue). The α3 helix of NS1 (red) is absent in M and is critical for IFN inhibition. NS2 (green) consists of a mixed alpha–beta fold that is unique. (PDB code for M: 4V23; PDB code for NS1: 5VJ2; PDB code for NS2: 7LDK). Figure prepared with PyMOL (http://www.pymol.org/ (accessed on 12 January 2022)).

**Figure 5 viruses-14-00419-f005:**
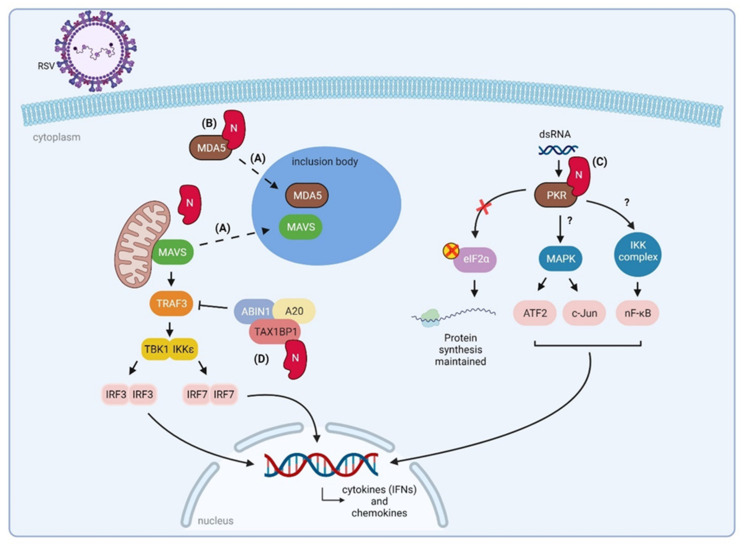
Proposed roles of RSV N in suppression of host innate immune responses. (**A**) RSV N sequesters MAVS and MDA5 to IBs. (**B**) RSV N directly interacts with MDA5. (**C**) RSV infection results in activation of PKR, which is involved in several signaling pathways. The binding of RSV N to PKR prevents phosphorylation of eIF2α by PKR and consequently to the conservation of mRNA translation. The biological effect of the RSV N-PKR interaction on downstream MAPK and IKK signaling remains elusive. (**D**) RSV N interacts with TAX1BP1, which is part of a multiprotein complex (together with ABIN1 and A20) that negatively regulates the host innate immune response by disrupting the interaction between TRAF3 and TBK1/IKKε. ABIN1, A20-binding inhibitor of NF-κB1; ATF2, activating transcription factor 2; eIF2α, α subunit of eukaryotic translation initiation factor 2; IBs, inclusion bodies; IKK, inhibitor of nuclear factor kappa-B kinase; IRF3, interferon regulatory factor 3; IRF7, interferon regulatory factor 7; MAVS, mitochondrial antiviral-signaling protein; MDA5, melanoma differentiation-associated protein 5; TBK1, tank binding kinase 1; TAX1BP1, Tax1-binding protein 1. Figure created with BioRender software (BioRender.com (accessed on 13 January 2022)).

**Figure 6 viruses-14-00419-f006:**
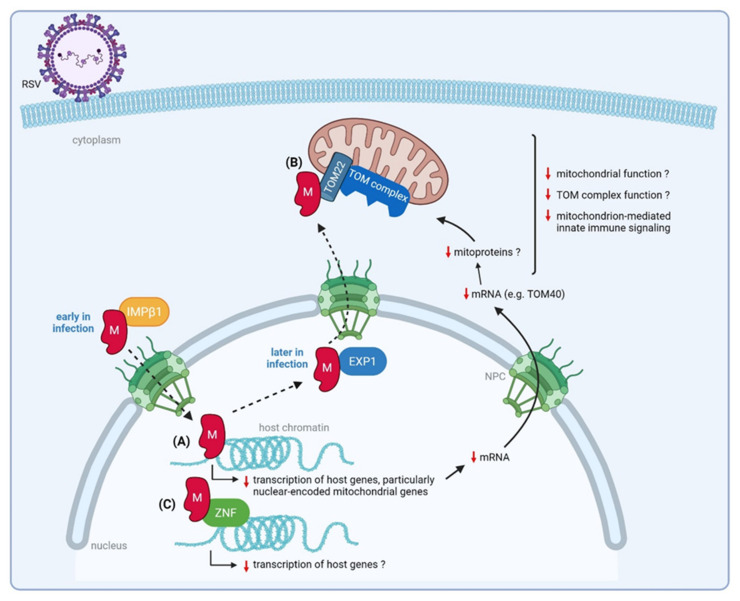
Proposed roles of RSV M in suppression of host innate immune responses. Early in infection, M translocates to the host nucleus. (**A**) Nuclear RSV M associates with host chromatin, resulting in inhibition of host gene transcription, in particular nuclear-encoded mitochondrial genes. Consequently, the function of the mitochondria and the TOM complex might be impaired, resulting in a decreased mitochondrion-mediated innate immune response. (**B**) RSV M directly interacts with TOM22, a subunit of the TOM complex. Since this complex has an antiviral role during RSV infection, probably through recruitment of innate immune factors to the mitochondria, the interaction of RSV M with a TOM subunit might lead to subversion of the host innate immune response, favoring virus production. (**C**) Nuclear RSV M interacts with ZNFs (as shown for ZNF502), possibly leading to impaired transcription of host genes. NPC, nuclear pore complex; IMPβ1, importin β1; EXP1, exportin 1; ZNF, Zinc-finger proteins. Figure created with BioRender software (BioRender.com (accessed on 13 January 2022)).

**Figure 7 viruses-14-00419-f007:**
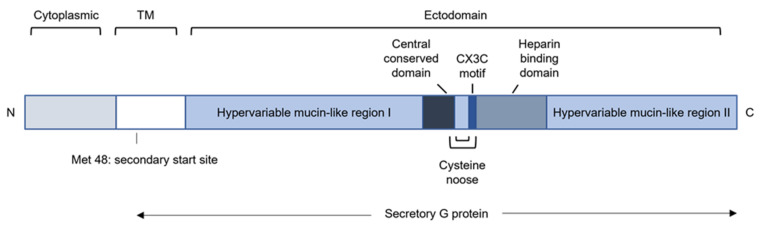
Schematic representation of the RSV G protein. The cytoplasmic and transmembrane (TM) domains of mG are localized at the N-terminus. The ectodomain includes two mucin-like regions that are separated by a central conserved domain that forms 2 disulfide bridges with the adjacent region, resembling a cysteine noose. The CX3C motif and heparin-binding domain are also part of the large ectodomain. Synthesis of secretory G protein occurs through translation initiation at the second start codon at position 48.

## Data Availability

Not applicable.

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
