# Peer review of "How RSV Proteins Join Forces to Overcome the Host Innate Immune Response"

_viruses, 2022, doi:10.3390/v14020419_

Round 1

Reviewer 1 Report

Infection with RSV is typically associated with low to undetectable levels of type I interferons because multiple RSV proteins can hinder the host's innate immune response. This review article summarizes the effects of various RSV proteins on the expression of interferons. This manuscript is well organized and described clearly.

Several minor suggestions:

  1. Figure legend 1. It is better to mention the length of the viral genome and the size of viral proteins.
  2. Lines 156, 369, empty space in the front of the first sentence.
  3. Lines 177-192, there is a reference for M protein but not for other viral proteins. It is better to have references after these viral proteins.
  4. Lines 227-231, there should be a reference for this section.
  5. Line 625, there should be a reference after [Several reports have highlighted that G, in addition to its attachment function, may contribute to RSV evasion of host immunity in a variety of ways].

Reviewer 2 Report

In this review, the authors describe the role of Respiratory syncytial virus proteins in host innate immune evasion. Overall the idea of this review seems to be curious, the paper is fairly written  and numerous figures are commendable,  however, I would have some suggestions to improve it for better readability.

Major issue

  • The manuscript could be better organized. In introduction there is no general information on which innate mechanisms are important in anti-viral defense and how they decrease the viral load. For example there is no introduction on which enzymes are activated in cells in IFN-I-dependent manner and what are they role to prevent virus infection. This is important because later in the manuscript these enzymes e.g PKR are targeted by RSV. The authors should remember that review article is read not only by microbiologists and immunobiologists but also by scientists from other disciplines, and, what is extremely important, by students.
  • The individual subchapters are very long. This unreadability makes that the reader needs to concentrate or guess more on how things are being said rather than what is being said.  Especially for a review article, this needs to be remedied. Subchapters should definitely be divided into sections explaining the general function of described protein (similarly as it was done to the subchapter 3.5) to make it easier to find your way around the issues described.
  • One paragraph does not follow an organized structure and is unfocused. The section 3.5.3 is definitely dedicated to humoral adaptive immunity based of neutralizing antibodies and effector functions mediated by these glycoproteins. If the authors want to live this section it should be reorganized to emphasize in more details the role of innate cells participating in antibody-mediated mechanisms.

Minor issue:

  • There are incorrect or at least incomprehensible sentences: Lines 645-649 – neutrophils and other granulocytes are not involved in clearance of virus-infected cells !
  • Line 645: … should be RSV not RSVDsG
